# Retrieval-Augmented Generation for Knowledge-Intensive NLP Tasks

**Patrick Lewis[†‡], Ethan Perez[⋆],**

**Aleksandra Piktus[†], Fabio Petroni[†], Vladimir Karpukhin[†], Naman Goyal[†], Heinrich Küttler[†],**

**Mike Lewis[†], Wen-tau Yih[†], Tim Rocktäschel[†‡], Sebastian Riedel[†‡], Douwe Kiela[†]**

[†]Facebook AI Research; [‡]University College London; [⋆]New York University;
`plewis@fb.com`

## Abstract

Large pre-trained language models have been shown to store factual knowledge in their parameters, and achieve state-of-the-art results when fine-tuned on downstream NLP tasks. However, their ability to access and precisely manipulate knowledge is still limited, and hence on knowledge-intensive tasks, their performance lags behind task-specific architectures. Additionally, providing provenance for their decisions and updating their world knowledge remain open research problems. Pre-trained models with a differentiable access mechanism to explicit non-parametric memory can overcome this issue, but have so far been only investigated for extractive downstream tasks. We explore a general-purpose fine-tuning recipe for retrieval-augmented generation (RAG) — models which combine pre-trained parametric and non-parametric memory for language generation. We introduce RAG models where the parametric memory is a pre-trained seq2seq model and the non-parametric memory is a dense vector index of Wikipedia, accessed with a pre-trained neural retriever. We compare two RAG formulations, one which conditions on the same retrieved passages across the whole generated sequence, and another which can use different passages per token. We fine-tune and evaluate our models on a wide range of knowledge-intensive NLP tasks and set the state of the art on three open domain QA tasks, outperforming parametric seq2seq models and task-specific retrieve-and-extract architectures. For language generation tasks, we find that RAG models generate more specific, diverse and factual language than a state-of-the-art parametric-only seq2seq baseline.

## 1 Introduction

Pre-trained neural language models have been shown to learn a substantial amount of in-depth knowledge from data [47]. They can do so without any access to an external memory, as a parameterized implicit knowledge base [51, 52]. While this development is exciting, such models do have downsides: They cannot easily expand or revise their memory, can't straightforwardly provide insight into their predictions, and may produce "hallucinations" [38]. Hybrid models that combine parametric memory with non-parametric (i.e., retrieval-based) memories [20, 26, 48] can address some of these issues because knowledge can be directly revised and expanded, and accessed knowledge can be inspected and interpreted. REALM [20] and ORQA [31], two recently introduced models that combine masked language models [8] with a differentiable retriever, have shown promising results,

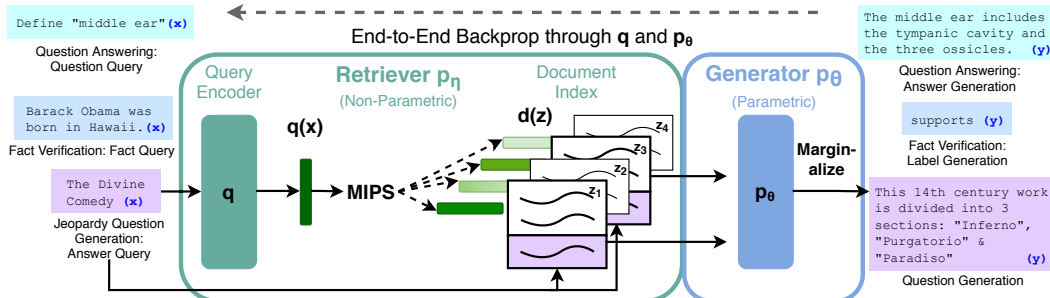

Figure 1: Overview of our approach. We combine a pre-trained retriever (*Query Encoder + Document Index*) with a pre-trained seq2seq model (*Generator*) and fine-tune end-to-end. For query $x$, we use Maximum Inner Product Search (MIPS) to find the top-K documents $z_i$. For final prediction $y$, we treat $z$ as a latent variable and marginalize over seq2seq predictions given different documents.

but have only explored open-domain extractive question answering. Here, we bring hybrid parametric and non-parametric memory to the "workhorse of NLP," i.e. sequence-to-sequence (seq2seq) models.

We endow pre-trained, parametric-memory generation models with a non-parametric memory through a general-purpose fine-tuning approach which we refer to as retrieval-augmented generation (RAG). We build RAG models where the parametric memory is a pre-trained seq2seq transformer, and the non-parametric memory is a dense vector index of Wikipedia, accessed with a pre-trained neural retriever. We combine these components in a probabilistic model trained end-to-end (Fig. 1). The retriever (Dense Passage Retriever [26], henceforth DPR) provides latent documents conditioned on the input, and the seq2seq model (BART [32]) then conditions on these latent documents together with the input to generate the output. We marginalize the latent documents with a top-K approximation, either on a per-output basis (assuming the same document is responsible for all tokens) or a per-token basis (where different documents are responsible for different tokens). Like T5 [51] or BART, RAG can be fine-tuned on any seq2seq task, whereby both the generator and retriever are jointly learned.

There has been extensive previous work proposing architectures to enrich systems with non-parametric memory which are trained from scratch for specific tasks, e.g. memory networks [64, 55], stack-augmented networks [25] and memory layers [30]. In contrast, we explore a setting where both parametric and non-parametric memory components are pre-trained and pre-loaded with extensive knowledge. Crucially, by using pre-trained access mechanisms, the ability to access knowledge is present without additional training.

Our results highlight the benefits of combining parametric and non-parametric memory with genera-tion for *knowledge-intensive tasks*—tasks that humans could not reasonably be expected to perform without access to an external knowledge source. Our RAG models achieve state-of-the-art results on open Natural Questions [29], WebQuestions [3] and CuratedTrec [2] and strongly outperform recent approaches that use specialised pre-training objectives on TriviaQA [24]. Despite these being extractive tasks, we find that unconstrained generation outperforms previous extractive approaches. For knowledge-intensive generation, we experiment with MS-MARCO [1] and Jeopardy question generation, and we find that our models generate responses that are more factual, specific, and diverse than a BART baseline. For FEVER [56] fact verification, we achieve results within 4.3% of state-of-the-art pipeline models which use strong retrieval supervision. Finally, we demonstrate that the non-parametric memory can be replaced to update the models' knowledge as the world changes.[1]

## 2 Methods

We explore RAG models, which use the input sequence $x$ to retrieve text documents $z$ and use them as additional context when generating the target sequence $y$. As shown in Figure 1, our models leverage two components: (i) a retriever $p_\eta(z|x)$ with parameters $\eta$ that returns (top-K truncated) distributions over text passages given a query $x$ and (ii) a generator $p_\theta(y_i|x, z, y_{1:i-1})$ parametrized

by $\theta$ that generates a current token based on a context of the previous $i-1$ tokens $y_{1:i-1}$, the original input $x$ and a retrieved passage $z$.

To train the retriever and generator end-to-end, we treat the retrieved document as a latent variable. We propose two models that marginalize over the latent documents in different ways to produce a distribution over generated text. In one approach, *RAG-Sequence*, the model uses the same document to predict each target token. The second approach, *RAG-Token*, can predict each target token based on a different document. In the following, we formally introduce both models and then describe the $p_\eta$ and $p_\theta$ components, as well as the training and decoding procedure.

## 2.1 Models

**RAG-Sequence Model**    The RAG-Sequence model uses the same retrieved document to generate the complete *sequence*. Technically, it treats the retrieved document as a single latent variable that is marginalized to get the seq2seq probability $p(y|x)$ via a top-K approximation. Concretely, the top K documents are retrieved using the retriever, and the generator produces the output sequence probability for each document, which are then marginalized,

$$p_{\text{RAG-Sequence}}(y|x) \approx \sum_{z \in \text{top-}k(p(\cdot|x))} p_\eta(z|x) p_\theta(y|x,z) = \sum_{z \in \text{top-}k(p(\cdot|x))} p_\eta(z|x) \prod_i^N p_\theta(y_i|x,z,y_{1:i-1})$$

**RAG-Token Model**    In the RAG-Token model we can draw a different latent document for each target *token* and marginalize accordingly. This allows the generator to choose content from several documents when producing an answer. Concretely, the top K documents are retrieved using the retriever, and then the generator produces a distribution for the next output token for each document, before marginalizing, and repeating the process with the following output token, Formally, we define:

$$p_{\text{RAG-Token}}(y|x) \approx \prod_i^N \sum_{z \in \text{top-}k(p(\cdot|x))} p_\eta(z|x) p_\theta(y_i|x,z_i,y_{1:i-1})$$

Finally, we note that RAG can be used for sequence classification tasks by considering the target class as a target sequence of length one, in which case RAG-Sequence and RAG-Token are equivalent.

## 2.2 Retriever: DPR

The retrieval component $p_\eta(z|x)$ is based on DPR [26]. DPR follows a bi-encoder architecture:

$$p_\eta(z|x) \propto \exp\left(\mathbf{d}(z)^\top \mathbf{q}(x)\right) \qquad \mathbf{d}(z) = \text{BERT}_d(z), \ \ \mathbf{q}(x) = \text{BERT}_q(x)$$

where $\mathbf{d}(z)$ is a dense representation of a document produced by a $\text{BERT}_{\text{BASE}}$ *document encoder* [8], and $\mathbf{q}(x)$ a query representation produced by a *query encoder*, also based on $\text{BERT}_{\text{BASE}}$. Calculating top-$k(p_\eta(\cdot|x))$, the list of $k$ documents $z$ with highest prior probability $p_\eta(z|x)$, is a Maximum Inner Product Search (MIPS) problem, which can be approximately solved in sub-linear time [23]. We use a pre-trained bi-encoder from DPR to initialize our retriever and to build the document index. This retriever was trained to retrieve documents which contain answers to TriviaQA [24] questions and Natural Questions [29]. We refer to the document index as the *non-parametric memory*.

## 2.3 Generator: BART

The generator component $p_\theta(y_i|x,z,y_{1:i-1})$ could be modelled using any encoder-decoder. We use BART-large [32], a pre-trained seq2seq transformer [58] with 400M parameters. To combine the input $x$ with the retrieved content $z$ when generating from BART, we simply concatenate them. BART was pre-trained using a denoising objective and a variety of different noising functions. It has obtained state-of-the-art results on a diverse set of generation tasks and outperforms comparably-sized T5 models [32]. We refer to the BART generator parameters $\theta$ as the *parametric memory* henceforth.

## 2.4 Training

We jointly train the retriever and generator components without any direct supervision on what document should be retrieved. Given a fine-tuning training corpus of input/output pairs $(x_j, y_j)$, we

minimize the negative marginal log-likelihood of each target, $\sum_j - \log p(y_j|x_j)$ using stochastic gradient descent with Adam [28]. Updating the document encoder $\text{BERT}_d$ during training is costly as it requires the document index to be periodically updated as REALM does during pre-training [20]. We do not find this step necessary for strong performance, and keep the document encoder (and index) fixed, only fine-tuning the query encoder $\text{BERT}_q$ and the BART generator.

## 2.5 Decoding

At test time, RAG-Sequence and RAG-Token require different ways to approximate $\arg\max_y p(y|x)$.

**RAG-Token** The RAG-Token model can be seen as a standard, autoregressive seq2seq genera-tor with transition probability: $p'_\theta(y_i|x, y_{1:i-1}) = \sum_{z \in \text{top-}k(p(\cdot|x))} p_\eta(z_i|x) p_\theta(y_i|x, z_i, y_{1:i-1})$ To decode, we can plug $p'_\theta(y_i|x, y_{1:i-1})$ into a standard beam decoder.

**RAG-Sequence** For RAG-Sequence, the likelihood $p(y|x)$ does not break into a conventional per-token likelihood, hence we cannot solve it with a single beam search. Instead, we run beam search for each document $z$, scoring each hypothesis using $p_\theta(y_i|x, z, y_{1:i-1})$. This yields a set of hypotheses $Y$, some of which may not have appeared in the beams of all documents. To estimate the probability of an hypothesis $y$ we run an additional forward pass for each document $z$ for which $y$ does not appear in the beam, multiply generator probability with $p_\eta(z|x)$ and then sum the probabilities across beams for the marginals. We refer to this decoding procedure as "Thorough Decoding." For longer output sequences, $|Y|$ can become large, requiring many forward passes. For more efficient decoding, we can make a further approximation that $p_\theta(y|x, z_i) \approx 0$ where $y$ was not generated during beam search from $x, z_i$. This avoids the need to run additional forward passes once the candidate set $Y$ has been generated. We refer to this decoding procedure as "Fast Decoding."

## 3 Experiments

We experiment with RAG in a wide range of knowledge-intensive tasks. For all experiments, we use a single Wikipedia dump for our non-parametric knowledge source. Following Lee et al. [31] and Karpukhin et al. [26], we use the December 2018 dump. Each Wikipedia article is split into disjoint 100-word chunks, to make a total of 21M documents. We use the document encoder to compute an embedding for each document, and build a single MIPS index using FAISS [23] with a Hierarchical Navigable Small World approximation for fast retrieval [37]. During training, we retrieve the top $k$ documents for each query. We consider $k \in \{5, 10\}$ for training and set $k$ for test time using dev data. We now discuss experimental details for each task.

### 3.1 Open-domain Question Answering

Open-domain question answering (QA) is an important real-world application and common testbed for knowledge-intensive tasks [20]. We treat questions and answers as input-output text pairs $(x, y)$ and train RAG by directly minimizing the negative log-likelihood of answers. We compare RAG to the popular extractive QA paradigm [5, 7, 31, 26], where answers are extracted spans from retrieved documents, relying primarily on non-parametric knowledge. We also compare to "Closed-Book QA" approaches [52], which, like RAG, generate answers, but which do not exploit retrieval, instead relying purely on parametric knowledge. We consider four popular open-domain QA datasets: Natural Questions (NQ) [29], TriviaQA (TQA) [24]. WebQuestions (WQ) [3] and CuratedTrec (CT) [2]. As CT and WQ are small, we follow DPR [26] by initializing CT and WQ models with our NQ RAG model. We use the same train/dev/test splits as prior work [31, 26] and report Exact Match (EM) scores. For TQA, to compare with T5 [52], we also evaluate on the TQA Wiki test set.

### 3.2 Abstractive Question Answering

RAG models can go beyond simple extractive QA and answer questions with free-form, abstractive text generation. To test RAG's natural language generation (NLG) in a knowledge-intensive setting, we use the MSMARCO NLG task v2.1 [43]. The task consists of questions, ten gold passages retrieved from a search engine for each question, and a full sentence answer annotated from the retrieved passages. We do not use the supplied passages, only the questions and answers, to treat

MSMARCO as an open-domain abstractive QA task. MSMARCO has some questions that cannot be answered in a way that matches the reference answer without access to the gold passages, such as "What is the weather in Volcano, CA?" so performance will be lower without using gold passages. We also note that some MSMARCO questions cannot be answered using Wikipedia alone. Here, RAG can rely on parametric knowledge to generate reasonable responses.

### 3.3 Jeopardy Question Generation

To evaluate RAG's generation abilities in a non-QA setting, we study open-domain question generation. Rather than use questions from standard open-domain QA tasks, which typically consist of short, simple questions, we propose the more demanding task of generating Jeopardy questions. Jeopardy is an unusual format that consists of trying to guess an entity from a fact about that entity. For example, "The World Cup" is the answer to the question "In 1986 Mexico scored as the first country to host this international sports competition twice." As Jeopardy questions are precise, factual statements, generating Jeopardy questions conditioned on their answer entities constitutes a challenging knowledge-intensive generation task.

We use the splits from SearchQA [10], with 100K train, 14K dev, and 27K test examples. As this is a new task, we train a BART model for comparison. Following [67], we evaluate using the SQuAD-tuned Q-BLEU-1 metric [42]. Q-BLEU is a variant of BLEU with a higher weight for matching entities and has higher correlation with human judgment for question generation than standard metrics. We also perform two human evaluations, one to assess generation factuality, and one for specificity. We define factuality as whether a statement can be corroborated by trusted external sources, and specificity as high mutual dependence between the input and output [33]. We follow best practice and use pairwise comparative evaluation [34]. Evaluators are shown an answer and two generated questions, one from BART and one from RAG. They are then asked to pick one of four options—quuestion A is better, question B is better, both are good, or neither is good.

### 3.4 Fact Verification

FEVER [56] requires classifying whether a natural language claim is supported or refuted by Wikipedia, or whether there is not enough information to decide. The task requires retrieving evidence from Wikipedia relating to the claim and then reasoning over this evidence to classify whether the claim is true, false, or unverifiable from Wikipedia alone. FEVER is a retrieval problem coupled with an challenging entailment reasoning task. It also provides an appropriate testbed for exploring the RAG models' ability to handle classification rather than generation. We map FEVER class labels (supports, refutes, or not enough info) to single output tokens and directly train with claim-class pairs. Crucially, unlike most other approaches to FEVER, we do not use supervision on retrieved evidence. In many real-world applications, retrieval supervision signals aren't available, and models that do not require such supervision will be applicable to a wider range of tasks. We explore two variants: the standard 3-way classification task (supports/refutes/not enough info) and the 2-way (supports/refutes) task studied in Thorne and Vlachos [57]. In both cases we report label accuracy.

## 4 Results

### 4.1 Open-domain Question Answering

Table 1 shows results for RAG along with state-of-the-art models. On all four open-domain QA tasks, RAG sets a new state of the art (only on the T5-comparable split for TQA). RAG combines the generation flexibility of the "closed-book" (parametric only) approaches and the performance of "open-book" retrieval-based approaches. Unlike REALM and T5+SSM, RAG enjoys strong results without expensive, specialized "salient span masking" pre-training [20]. It is worth noting that RAG's retriever is initialized using DPR's retriever, which uses retrieval supervision on Natural Questions and TriviaQA. RAG compares favourably to the DPR QA system, which uses a BERT-based "cross-encoder" to re-rank documents, along with an extractive reader. RAG demonstrates that neither a re-ranker nor extractive reader is necessary for state-of-the-art performance.

There are several advantages to generating answers even when it is possible to extract them. Documents with clues about the answer but do not contain the answer verbatim can still contribute towards a correct answer being generated, which is not possible with standard extractive approaches, leading

Table 1: Open-Domain QA Test Scores. For TQA, left column uses the standard test set for Open-Domain QA, right column uses the TQA-Wiki test set. See Appendix D for further details.

|  | Model | NQ | TQA | WQ | CT |
|---|---|---|---|---|---|
| Closed Book | T5-11B [52] | 34.5 | - /50.1 | 37.4 | - |
|  | T5-11B+SSM[52] | 36.6 | - /60.5 | 44.7 | - |
| Open Book | REALM [20] | 40.4 | - / - | 40.7 | 46.8 |
|  | DPR [26] | 41.5 | **57.9**/ - | 41.1 | 50.6 |
|  | RAG-Token | 44.1 | 55.2/66.1 | **45.5** | 50.0 |
|  | RAG-Seq. | **44.5** | 56.8/**68.0** | 45.2 | **52.2** |

Table 2: Generation and classification Test Scores. MS-MARCO SotA is [4], FEVER-3 is [68] and FEVER-2 is [57] *Uses gold context/evidence. Best model without gold access underlined.

| Model | Jeopardy B-1 | QB-1 | MSMARCO R-L | B-1 | FVR3 Label | FVR2 Acc. |
|---|---|---|---|---|---|---|
| SotA | - | - | **49.8*** | **49.9*** | **76.8** | **92.2*** |
| BART | 15.1 | 19.7 | 38.2 | 41.6 | 64.0 | 81.1 |
| RAG-Tok. | **17.3** | **22.2** | 40.1 | 41.5 | 72.5 | 89.5 |
| RAG-Seq. | 14.7 | 21.4 | 40.8 | 44.2 |  | 89.5 |

to more effective marginalization over documents. Furthermore, RAG can generate correct answers even when the correct answer is not in any retrieved document, achieving 11.8% accuracy in such cases for NQ, where an extractive model would score 0%.

## 4.2 Abstractive Question Answering

As shown in Table 2, RAG-Sequence outperforms BART on Open MS-MARCO NLG by 2.6 Bleu points and 2.6 Rouge-L points. RAG approaches state-of-the-art model performance, which is impressive given that (i) those models access gold passages with specific information required to generate the reference answer, (ii) many questions are unanswerable without the gold passages, and (iii) not all questions are answerable from Wikipedia alone. Table 3 shows some generated answers from our models. Qualitatively, we find that RAG models hallucinate less and generate factually correct text more often than BART. Later, we also show that RAG generations are more diverse than BART generations (see §4.5).

## 4.3 Jeopardy Question Generation

Table 2 shows that RAG-Token performs better than RAG-Sequence on Jeopardy question generation, with both models outperforming BART on Q-BLEU-1. Table 4 shows human evaluation results, over 452 pairs of generations from BART and RAG-Token. Evaluators indicated that BART was more factual than RAG in only 7.1% of cases, while RAG was more factual in 42.7% of cases, and both RAG and BART were factual in a further 17% of cases, clearly demonstrating the effectiveness of RAG on the task over a state-of-the-art generation model. Evaluators also find RAG generations to be more specific by a large margin. Table 3 shows typical generations from each model.

Jeopardy questions often contain two separate pieces of information, and RAG-Token may perform best because it can generate responses that combine content from several documents. Figure 2 shows an example. When generating "Sun", the posterior is high for document 2 which mentions "The Sun Also Rises". Similarly, document 1 dominates the posterior when "A Farewell to Arms" is generated. Intriguingly, after the first token of each book is generated, the document posterior flattens. This observation suggests that the generator can complete the titles without depending on specific documents. In other words, the model's parametric knowledge is sufficient to complete the titles. We find evidence for this hypothesis by feeding the BART-only baseline with the partial decoding `"The Sun`. BART completes the generation `"The Sun Also Rises" is a novel by this author of "The Sun Also Rises"` indicating the title "The Sun Also Rises" is stored in BART's parameters. Similarly, BART will complete the partial decoding `"The Sun Also Rises" is a novel by this author of "A` with `"The Sun Also Rises" is a novel by this author of "A Farewell to Arms"`. This example shows how parametric and non-parametric memories *work together*—the non-parametric component helps to guide the generation, drawing out specific knowledge stored in the parametric memory.

## 4.4 Fact Verification

Table 2 shows our results on FEVER. For 3-way classification, RAG scores are within 4.3% of state-of-the-art models, which are complex pipeline systems with domain-specific architectures and substantial engineering, trained using intermediate retrieval supervision, which RAG does not require.

**Document 1**: his works are considered classics of American literature ... His wartime experiences formed the basis for his novel **"A Farewell to Arms"** (1929) ...

**Document 2**: ... artists of the 1920s "Lost Generation" expatriate community. His debut novel, **"The Sun Also Rises"**, was published in 1926.

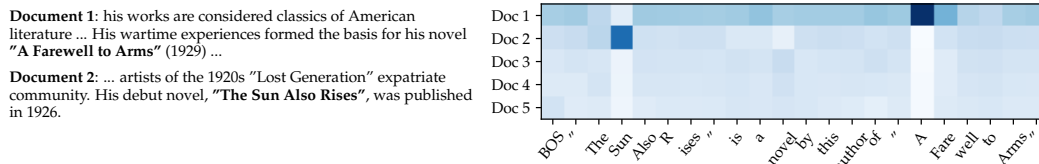

Figure 2: RAG-Token document posterior $p(z_i|x, y_i, y_{-i})$ for each generated token for input "Hemingway" for Jeopardy generation with 5 retrieved documents. The posterior for document 1 is high when generating "A Farewell to Arms" and for document 2 when generating "The Sun Also Rises".

Table 3: Examples from generation tasks. RAG models generate more specific and factually accurate responses. '?' indicates factually incorrect responses, * indicates partially correct responses.

| Task | Input | Model | Generation |
|---|---|---|---|
| MS-MARCO | define middle ear | BART | ?The middle ear is the part of the ear between the middle ear and the nose. |
| | | RAG-T | The middle ear is the portion of the ear internal to the eardrum. |
| | | RAG-S | The middle ear includes the tympanic cavity and the three ossicles. |
| | what currency needed in scotland | BART | The currency needed in Scotland is Pound sterling. |
| | | RAG-T | Pound is the currency needed in Scotland. |
| | | RAG-S | The currency needed in Scotland is the pound sterling. |
| Jeopardy Question Gener-ation | Washington | BART | ?This state has the largest number of counties in the U.S. |
| | | RAG-T | It's the only U.S. state named for a U.S. president |
| | | RAG-S | It's the state where you'll find Mount Rainier National Park |
| | The Divine Comedy | BART | *This epic poem by Dante is divided into 3 parts: the Inferno, the Purgatorio & the Purgatorio |
| | | RAG-T | Dante's "Inferno" is the first part of this epic poem |
| | | RAG-S | This 14th century work is divided into 3 sections: "Inferno", "Purgatorio" & "Paradiso" |

For 2-way classification, we compare against Thorne and Vlachos [57], who train RoBERTa [35] to classify the claim as true or false given the gold evidence sentence. RAG achieves an accuracy within 2.7% of this model, despite being supplied with only the claim and retrieving its own evidence. We also analyze whether documents retrieved by RAG correspond to documents annotated as gold evidence in FEVER. We calculate the overlap in article titles between the top $k$ documents retrieved by RAG and gold evidence annotations. We find that the top retrieved document is from a gold article in 71% of cases, and a gold article is present in the top 10 retrieved articles in 90% of cases.

### 4.5 Additional Results

**Generation Diversity** Section 4.3 shows that RAG models are more factual and specific than BART for Jeopardy question generation. Following recent work on diversity-promoting decoding [33, 59, 39], we also investigate generation diversity by calculating the ratio of distinct ngrams to total ngrams generated by different models. Table 5 shows that RAG-Sequence's generations are more diverse than RAG-Token's, and both are significantly more diverse than BART without needing any diversity-promoting decoding.

**Retrieval Ablations** A key feature of RAG is learning to retrieve relevant information for the task. To assess the effectiveness of the retrieval mechanism, we run ablations where we freeze the retriever during training. As shown in Table 6, learned retrieval improves results for all tasks. We compare RAG's dense retriever to a word overlap-based BM25 retriever [53]. Here, we replace RAG's retriever with a fixed BM25 system, and use BM25 retrieval scores as logits when calculating $p(z|x)$. Table 6 show the results. For FEVER, BM25 performs best, perhaps since FEVER claims are heavily entity-centric and thus well-suited for word overlap-based retrieval. Differentiable retrieval improves results on all other tasks, especially for Open-Domain QA, where it is crucial.

**Index hot-swapping** An advantage of non-parametric memory models like RAG is that knowledge can be easily updated at test time. Parametric-only models like T5 or BART need further training to update their behavior as the world changes. To demonstrate, we build an index using the DrQA [5] Wikipedia dump from December 2016 and compare outputs from RAG using this index to the newer index from our main results (December 2018). We prepare a list of 82 world leaders who had changed between these dates and use a template "Who is {position}?" (e.g. "Who is the President of Peru?")

Table 4: Human assessments for the Jeopardy Question Generation Task.

|  | Factuality | Specificity |
|---|---|---|
| BART better | 7.1% | 16.8% |
| RAG better | **42.7%** | **37.4%** |
| Both good | 11.7% | 11.8% |
| Both poor | 17.7% | 6.9% |
| No majority | 20.8% | 20.1% |

Table 5: Ratio of distinct to total tri-grams for generation tasks.

|  | MSMARCO | Jeopardy QGen |
|---|---|---|
| Gold | 89.6% | 90.0% |
| BART | 70.7% | 32.4% |
| RAG-Token | 77.8% | 46.8% |
| RAG-Seq. | 83.5% | 53.8% |

Table 6: Ablations on the dev set. As FEVER is a classification task, both RAG models are equivalent.

| Model | NQ | TQA | WQ | CT | Jeopardy-QGen | | MSMarco | | FVR-3 | FVR-2 |
|---|---|---|---|---|---|---|---|---|---|---|
|  | Exact Match | | | | B-1 | QB-1 | R-L | B-1 | Label Accuracy | |
| RAG-Token-BM25 | 29.7 | 41.5 | 32.1 | 33.1 | 17.5 | 22.3 | 55.5 | 48.4 | **75.1** | **91.6** |
| RAG-Sequence-BM25 | 31.8 | 44.1 | 36.6 | 33.8 | 11.1 | 19.5 | 56.5 | 46.9 | | |
| RAG-Token-Frozen | 37.8 | 50.1 | 37.1 | 51.1 | 16.7 | 21.7 | 55.9 | 49.4 | 72.9 | 89.4 |
| RAG-Sequence-Frozen | 41.2 | 52.1 | 41.8 | 52.6 | 11.8 | 19.6 | 56.7 | 47.3 | | |
| RAG-Token | 43.5 | 54.8 | **46.5** | 51.9 | **17.9** | **22.6** | 56.2 | **49.4** | 74.5 | 90.6 |
| RAG-Sequence | **44.0** | **55.8** | 44.9 | **53.4** | 15.3 | 21.5 | **57.2** | 47.5 | | |

to query our NQ RAG model with each index. RAG answers 70% correctly using the 2016 index for 2016 world leaders and 68% using the 2018 index for 2018 world leaders. Accuracy with mismatched indices is low (12% with the 2018 index and 2016 leaders, 4% with the 2016 index and 2018 leaders). This shows we can update RAG's world knowledge by simply replacing its non-parametric memory.

**Effect of Retrieving more documents**  Models are trained with either 5 or 10 retrieved latent documents, and we do not observe significant differences in performance between them. We have the flexibility to adjust the number of retrieved documents at test time, which can affect performance and runtime. Figure 3 (left) shows that retrieving more documents at test time monotonically improves Open-domain QA results for RAG-Sequence, but performance peaks for RAG-Token at 10 retrieved documents. Figure 3 (right) shows that retrieving more documents leads to higher Rouge-L for RAG-Token at the expense of Bleu-1, but the effect is less pronounced for RAG-Sequence.

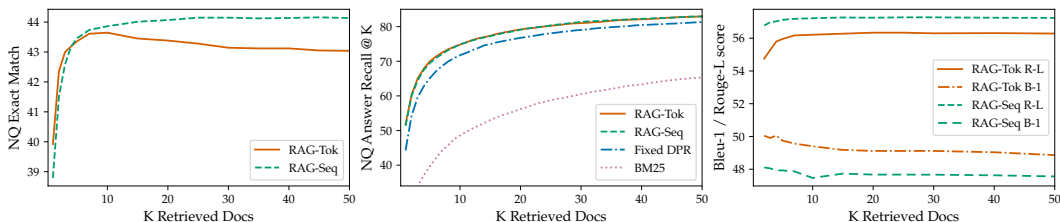

Figure 3: Left: NQ performance as more documents are retrieved. Center: Retrieval recall performance in NQ. Right: MS-MARCO Bleu-1 and Rouge-L as more documents are retrieved.

## 5 Related Work

**Single-Task Retrieval**  Prior work has shown that retrieval improves performance across a variety of NLP tasks when considered in isolation. Such tasks include open-domain question answering [5, 29], fact checking [56], fact completion [48], long-form question answering [12], Wikipedia article generation [36], dialogue [41, 65, 9, 13], translation [17], and language modeling [19, 27]. Our work unifies previous successes in incorporating retrieval into individual tasks, showing that a single retrieval-based architecture is capable of achieving strong performance across several tasks.

**General-Purpose Architectures for NLP**    Prior work on general-purpose architectures for NLP tasks has shown great success without the use of retrieval. A single, pre-trained language model has been shown to achieve strong performance on various classification tasks in the GLUE benchmarks [60, 61] after fine-tuning [49, 8]. GPT-2 [50] later showed that a single, left-to-right, pre-trained language model could achieve strong performance across both discriminative and generative tasks. For further improvement, BART [32] and T5 [51, 52] propose a single, pre-trained encoder-decoder model that leverages bi-directional attention to achieve stronger performance on discriminative and generative tasks. Our work aims to expand the space of possible tasks with a single, unified architecture, by learning a retrieval module to augment pre-trained, generative language models.

**Learned Retrieval**    There is significant work on learning to retrieve documents in information retrieval, more recently with pre-trained, neural language models [44, 26] similar to ours. Some work optimizes the retrieval module to aid in a specific, downstream task such as question answering, using search [46], reinforcement learning [6, 63, 62], or a latent variable approach [31, 20] as in our work. These successes leverage different retrieval-based architectures and optimization techniques to achieve strong performance on a single task, while we show that a single retrieval-based architecture can be fine-tuned for strong performance on a variety of tasks.

**Memory-based Architectures**    Our document index can be seen as a large external memory for neural networks to attend to, analogous to memory networks [64, 55]. Concurrent work [14] learns to retrieve a trained embedding for each entity in the input, rather than to retrieve raw text as in our work. Other work improves the ability of dialog models to generate factual text by attending over fact embeddings [9, 13] or, closer to our work, over retrieved text directly [15]. A key feature of our memory is that it is comprised of raw text rather distributed representations, which makes the memory both (i) human-readable, lending a form of interpretability to our model, and (ii) human-writable, enabling us to dynamically update the model's memory by editing the document index.

**Retrieve-and-Edit approaches**    Our method shares some similarities with retrieve-and-edit style approaches, where a similar training input-output pair is retrieved for a given input, and then edited to provide a final output. These approaches have proved successful in a number of domains including Machine Translation [18, 22] and Semantic Parsing [21]. Our approach does have several differences, including less of emphasis on lightly editing a retrieved item, but on aggregating content from several pieces of retrieved content, as well as learning latent retrieval, and retrieving evidence documents rather than related training pairs. This said, RAG techniques may work well in these settings, and could represent promising future work.

## 6    Discussion

In this work, we presented hybrid generation models with access to parametric and non-parametric memory. We showed that our RAG models obtain state of the art results on open-domain QA. We found that people prefer RAG's generation over purely parametric BART, finding RAG more factual and specific. We conducted an thorough investigation of the learned retrieval component, validating its effectiveness, and we illustrated how the retrieval index can be hot-swapped to update the model without requiring any retraining. In future work, it may be fruitful to investigate if the two components can be jointly pre-trained from scratch, either with a denoising objective similar to BART or some another objective. Our work opens up new research directions on how parametric and non-parametric memories interact and how to most effectively combine them, showing promise in being applied to a wide variety of NLP tasks.

## Broader Impact

This work offers several positive societal benefits over previous work: the fact that it is more strongly grounded in real factual knowledge (in this case Wikipedia) makes it "hallucinate" less with generations that are more factual, and offers more control and interpretability. RAG could be employed in a wide variety of scenarios with direct benefit to society, for example by endowing it with a medical index and asking it open-domain questions on that topic, or by helping people be more effective at their jobs.

With these advantages also come potential downsides: Wikipedia, or any potential external knowledge source, will probably never be entirely factual and completely devoid of bias. Since RAG can be employed as a language model, similar concerns as for GPT-2 [50] are valid here, although arguably to a lesser extent, including that it might be used to generate abuse, faked or misleading content in the news or on social media; to impersonate others; or to automate the production of spam/phishing content [54]. Advanced language models may also lead to the automation of various jobs in the coming decades [16]. In order to mitigate these risks, AI systems could be employed to fight against misleading content and automated spam/phishing.

## Acknowledgments

The authors would like to thank the reviewers for their thoughtful and constructive feedback on this paper, as well as HuggingFace for their help in open-sourcing code to run RAG models. The authors would also like to thank Kyunghyun Cho and Sewon Min for productive discussions and advice.

## Funding Disclosure

EP thanks supports from the NSF Graduate Research Fellowship. PL is supported by the FAIR PhD program. This work was funded by Facebook.

## Footnotes

[1]Code to run experiments with RAG has been open-sourced as part of the HuggingFace Transform-ers Library [66] and can be found at https://github.com/huggingface/transformers/blob/master/examples/rag/. An interactive demo of RAG models can be found at https://huggingface.co/rag/

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
