[Supplementary Material]

# Appendices for Retrieval-Augmented Generation for Knowledge-Intensive NLP Tasks

## A  Implementation Details

For Open-domain QA we report test numbers using 15 retrieved documents for RAG-Token models. For RAG-Sequence models, we report test results using 50 retrieved documents, and we use the Thorough Decoding approach since answers are generally short. We use greedy decoding for QA as we did not find beam search improved results. For Open-MSMarco and Jeopardy question generation, we report test numbers using ten retrieved documents for both RAG-Token and RAG-Sequence, and we also train a BART-large model as a baseline. We use a beam size of four, and use the Fast Decoding approach for RAG-Sequence models, as Thorough Decoding did not improve performance.

## B  Human Evaluation

Figure 4: Annotation interface for human evaluation of factuality. A pop-out for detailed instructions and a worked example appear when clicking "view tool guide".

Figure 4 shows the user interface for human evaluation. To avoid any biases for screen position, which model corresponded to sentence A and sentence B was randomly selected for each example. Annotators were encouraged to research the topic using the internet, and were given detailed instructions and worked examples in a full instructions tab. We included some gold sentences in order to assess the accuracy of the annotators. Two annotators did not perform well on these examples and their annotations were removed from the results.

## C  Training setup Details

We train all RAG models and BART baselines using Fairseq [45].[2] We train with mixed precision floating point arithmetic [40], distributing training across 8, 32GB NVIDIA V100 GPUs, though training and inference can be run on one GPU. We find that doing Maximum Inner Product Search with FAISS is sufficiently fast on CPU, so we store document index vectors on CPU, requiring $\sim 100$ GB of CPU memory for all of Wikipedia. After submission, We have ported our code to HuggingFace Transformers [66],[3] which achieves equivalent performance to the previous version but is a cleaner and easier to use implementation. This version is also open-sourced. We also compress the document index using FAISS's compression tools, reducing the CPU memory requirement to 36GB. Scripts to run experiments with RAG can be found at `https://github.com/huggingface/transformers/blob/master/examples/rag/README.md` and an interactive demo of a RAG model can be found at `https://huggingface.co/rag/`

# D   Further Details on Open-Domain QA

For open-domain QA, multiple answer annotations are often available for a given question. These answer annotations are exploited by extractive models during training as typically all the answer annotations are used to find matches within documents when preparing training data. For RAG, we also make use of multiple annotation examples for Natural Questions and WebQuestions by training the model with each $(q, a)$ pair separately, leading to a small increase in accuracy. For TriviaQA, there are often many valid answers to a given question, some of which are not suitable training targets, such as emoji or spelling variants. For TriviaQA, we filter out answer candidates if they do not occur in top 1000 documents for the query.

**CuratedTrec preprocessing**   The answers for CuratedTrec are given in the form of regular expressions, which has been suggested as a reason why it is unsuitable for answer-generation models [20]. To overcome this, we use a pre-processing step where we first retrieve the top 1000 documents for each query, and use the answer that most frequently matches the regex pattern as the supervision target. If no matches are found, we resort to a simple heuristic: generate all possible permutations for each regex, replacing non-deterministic symbols in the regex nested tree structure with a whitespace.

**TriviaQA Evaluation setups**   The open-domain QA community customarily uses public development datasets as test datasets, as test data for QA datasets is often restricted and dedicated to reading comprehension purposes. We report our results using the datasets splits used in DPR [26], which are consistent with common practice in Open-domain QA. For TriviaQA, this test dataset is the public TriviaQA Web Development split. Roberts et al. [52] used the TriviaQA official Wikipedia test set instead. Févry et al. [14] follow this convention in order to compare with Roberts et al. [52] (See appendix of [14]). We report results on both test sets to enable fair comparison to both approaches. We find that our performance is much higher using the official Wiki test set, rather than the more conventional open-domain test set, which we attribute to the official Wiki test set questions being simpler to answer from Wikipedia.

# E   Further Details on FEVER

For FEVER classification, we follow the practice from [32], and first re-generate the claim, and then classify using the representation of the final hidden state, before finally marginalizing across documents to obtain the class probabilities. The FEVER task traditionally has two sub-tasks. The first is to classify the claim as either "Supported", "Refuted" or "Not Enough Info", which is the task we explore in the main paper. FEVER's other sub-task involves extracting sentences from Wikipedia as evidence supporting the classification prediction. As FEVER uses a different Wikipedia dump to us, directly tackling this task is not straightforward. We hope to address this in future work.

# F   Null Document Probabilities

We experimented with adding "Null document" mechanism to RAG, similar to REALM [20] in order to model cases where no useful information could be retrieved for a given input. Here, if $k$ documents were retrieved, we would additionally "retrieve" an empty document and predict a logit for the null document, before marginalizing over $k + 1$ predictions. We explored modelling this null document logit by learning (i) a document embedding for the null document, (ii) a static learnt bias term, or (iii) a neural network to predict the logit. We did not find that these improved performance, so in the interests of simplicity, we omit them. For Open MS-MARCO, where useful retrieved documents cannot always be retrieved, we observe that the model learns to always retrieve a particular set of documents for questions that are less likely to benefit from retrieval, suggesting that null document mechanisms may not be necessary for RAG.

# G   Parameters

Our RAG models contain the trainable parameters for the BERT-base query and document encoder of DPR, with 110M parameters each (although we do not train the document encoder ourselves) and 406M trainable parameters from BART-large, 406M parameters, making a total of 626M trainable

Table 7: Number of instances in the datasets used. *A hidden subset of this data is used for evaluation

| Task | Train | Development | Test |
|---|---|---|---|
| Natural Questions | 79169 | 8758 | 3611 |
| TriviaQA | 78786 | 8838 | 11314 |
| WebQuestions | 3418 | 362 | 2033 |
| CuratedTrec | 635 | 134 | 635 |
| Jeopardy Question Generation | 97392 | 13714 | 26849 |
| MS-MARCO | 153726 | 12468 | 101093* |
| FEVER-3-way | 145450 | 10000 | 10000 |
| FEVER-2-way | 96966 | 6666 | 6666 |

parameters. The best performing "closed-book" (parametric only) open-domain QA model is T5-11B with 11 Billion trainable parameters. The T5 model with the closest number of parameters to our models is T5-large (770M parameters), which achieves a score of 28.9 EM on Natural Questions [52], substantially below the 44.5 that RAG-Sequence achieves, indicating that hybrid parametric/non-parametric models require far fewer trainable parameters for strong open-domain QA performance. The non-parametric memory index does not consist of trainable parameters, but does consists of 21M 728 dimensional vectors, consisting of 15.3B values. These can be easily be stored at 8-bit floating point precision to manage memory and disk footprints.

## H  Retrieval Collapse

In preliminary experiments, we observed that for some tasks such as story generation [11], the retrieval component would "collapse" and learn to retrieve the same documents regardless of the input. In these cases, once retrieval had collapsed, the generator would learn to ignore the documents, and the RAG model would perform equivalently to BART. The collapse could be due to a less-explicit requirement for factual knowledge in some tasks, or the longer target sequences, which could result in less informative gradients for the retriever. Perez et al. [46] also found spurious retrieval results when optimizing a retrieval component in order to improve performance on downstream tasks.

## I  Number of instances per dataset

The number of training, development and test datapoints in each of our datasets is shown in Table 7.

## Footnotes

[2]`https://github.com/pytorch/fairseq`

[3]`https://github.com/huggingface/transformers`