[Reviews · NeurIPS 2020]

Review 1

Summary and Contributions: This paper propose a hybrid generation models by integrating the information retrieval strategy (non-parametric memory) with seq2seq model(parametric memory). This model can work on more general knowledge-intensive tasks, such as QA, question generation and fact verification, other than only extractive downstream tasks. It outperforms seq2seq models and task-specific retrieve-and-extract models on three open domain QA tasks. In generation tasks, it can generate more specific, diverse and factual language than a seq2seq baseline model.

Strengths: 1.The motivation of using the the information retrieval strategy to improve the the generation models is sound. The paper did extensive experiments on several Knowledge-Intensive NLP Tasks, which show the effectiveness of the proposed approach.

Weaknesses: Using the the information retrieval strategy to improve the the generation models is straightforward and certainly not novel .Although, the paper did extensive experiments on several Knowledge-Intensive NLP Tasks and showed the effectiveness of the proposed approach. The contribution of the model is not very specific, since that RAG and its components are not proposed by authors.

Correctness: yes

Clarity: 1.A figure or example about PAG-Sequence Model and PAG-Token Model is needed, which can help people understand more about the difference between these two models. 2.The reason why it need not fine tune document encoder is not clear, which needs experimental result to support it.

Relation to Prior Work: yes

Reproducibility: No

Additional Feedback:


Review 2

Summary and Contributions: This paper proposes a retrieval-augmented generation framework to enhance the performance of NLP tasks required knowledge. A pre-trained document retriever (DPR) is introduced to conduct the query understanding and document retrieval. A pre-trained seq2seq encoder decoder framework (BART) is employed to conduct the context (query + retrieved documents) understanding and generation. Two information exploiting strategies are designed for the decoder. Experimental results show that the proposed framework can obtain better performance on the tasks of knowledge depended tasks.

Strengths: The proposed idea and approach are interesting, and the experimental results are also very good. The paper is in well-written and easy to follow. We believe that the retrieval+generation framework can be extended to more tasks such as machine translation and dialogue system.

Weaknesses: One tiny weakness is that the core technical components are just borrowed directly from the existing works, such as DPR and BART. And retrieval+extraction/generation framework have also been conducted by some previous works such as Danqi Chen’s work.

Correctness: yes

Clarity: yes

Relation to Prior Work: yes

Reproducibility: Yes

Additional Feedback: This is an interesting work and can benefit the community, and it should be accepted and introduced to public (though the pre-trained version is on arxiv).


Review 3

Summary and Contributions: This paper presents a new model for NLP tasks that require the use of external knowledge. In particular, the proposed approach combines a non-parametric memory (a dense vector index of Wikipedia) and a parametric memory (a pre-trained seq2seq transformer) to enrich the input query and address the considered task. Experiments conducted on a wide range of different tasks confirm the significance of the proposed approach. Although the method was primarily designed for open-domain question answering, it show nice results also on related NLP tasks, such as fact checking. *** AFTER THE REBUTTAL *** I thank the authors for the answers provided in the rebuttal.

Strengths: + Novelty of the approach + Extensive experimental evaluation + Relevance to the NeurIPS community

Weaknesses: - The description of the model is quite concise (due to space restrictions) - Reproducibility of the results (link to code and models is hidden for double blind review, so it is hard to judge reproducibility)

Correctness: The methodology is solid and correct

Clarity: The paper is very well written, just concise in the part of model description

Relation to Prior Work: The paper is well positioned against the state-of-the-art, but more experimental comparison would have strengthened the paper

Reproducibility: No

Additional Feedback: * Although the experimental section is very rich, I wonder whether it would have been possible to include more comparisons with some memory-based neural network. For example, one could think of applying a generator on top of a standard Memory-Augmented Neural Network, or some other similar approach. * In Section 2.4, it is stated that the model is trained "without any direct supervision on what document should be retrieved". I wonder whether it would be possible also to train the model with such an information, in the same way memory networks use strong supervision. Typos: - "can't" -> "cannot" - In footnote 1 there is a repetition of "available" - Pag. 4, "answers are extracted spans from retrieved documents" -> remove "spans"? - Pag. 5, "quuestion" -> "question"


Review 4

Summary and Contributions: This paper proposes a retrieval augmented seq2seq model for question answering and related knowledge-intensive NLP tasks. The model is combination of a pre-trained BART and a dense passage retriever via joint probabilistic model. Two specific formulations, referred to as RAG-Sequence and RAG-Token, are proposed to let the model select relevant document(s) to generate answers. Experiments are conducted on a range of tasks including open-domain question answering and fact verification, showing that the RAG model achieve state-of-the-art or competitive performances. The design of the model share some similarity with REALM model, which is also a retrieval augmented encoder-only model. REALM can be pre-trained on unsupervised documents and then fine-tuned for extractive question answering tasks. The RAG model proposed in this paper have an encoder-decoder structure which makes the model more flexible to handle a broader range of tasks. However, the retriever and seq2seq module of RAG model are pre-trained in a separated manner and only joint learned in the fine-tuning stage.

Strengths: The RAG model proposed in this paper benefits from both the parametric and non-parametric model and provide a promising framework for the open-domain question answering and knowledge-related NLP tasks. Although the idea may not be totally novel, it is the first model that integrates seq2seq pre-trained model with the neural retrieval model, which can be trained in an end-to-end fashion. The performance of the model is strong on several question answering tasks, proving it as a general framework for these tasks. This paper should be of great interest to the NeruIPS community, especially for researchers in NLP and IR field.

Weaknesses: The whole framework is in some sense the encoder-decoder version of the REALM model, while the two components (BART and DPR) cannot be pre-trained end-to-end, which prevent their synergy in learning the knowledge in the pre-training stage. For generation tasks, it would be more interesting to compare RAG with GPT-2 or T5 models rather than BART.

Correctness: Yes

Clarity: The paper is well organized and easy to follow.

Relation to Prior Work: Yes

Reproducibility: Yes

Additional Feedback:

[Author Response · NeurIPS 2020]

# NeurIPS Rebuttal for "Retrieval-Augmented Generation for Knowledge-Intensive NLP Tasks"

We thank reviewers for their thoughtful, detailed reviews. We shall discuss three common themes, before addressing individual comments.

**Novelty of retrieval-augmentation (R1, R2, R4)**  "information retrieval strategy to improve the the generation models is... not novel" (R1), "retrieval+extraction/generation framework have also been conducted by some previous works" (R2), and "the framework is in some sense the encoder-decoder version of the REALM model" (R4). We agree that retrieval is a well-known strategy in pipeline-based NLP systems. We emphasize the following points that we consider to be novel contributions in this area:

- Pre-trained seq2seq models have only become available in the last year (T5, BART) or two (GPT2). To our knowledge, work has not yet been published investigating how retrieval-augmentation affects this class of pretrained generators.
- Unlike prior generation work, we show retrieval and generation can be trained jointly with a single objective, demonstrating joint training is effective, outperforming both fixed retrieval and BM25 pipelines
- We study two RAG models. RAG-Sequence's formulation is similar to REALM, but RAG-Token is novel and has not been previously proposed. Further, we explore novel decoding strategies for these models.
- We demonstrate an effective general trainable retrieval-augmentation framework for *any* knowledge-intensive task, rather than ad-hoc task-specific architectures, such as Chen et al.'s DrQA. We feel this constitutes a novel and important class of model which combines parametric and non-parametric memories for any task.

**Reusing existing Components vs developing a novel model (R1, R2)**  "contribution [...] is not very specific, since that RAG and its components are not proposed by authors" (R1), and "One tiny weakness is that the core technical components are just borrowed directly from the existing works" (R2). We believe the ability to re-use existing components is specifically a strength of our framework:

- We believe avoiding pre-training is positive, as this saves computational resources and makes the process accessible to smaller labs. Re-using existing components is attractive in this respect and results in state-of-the-art performance, outperforming several approaches including end2end pretrained models.
- RAG is modular and agnostic to the retriever and generator, so it can directly benefit from future innovations in generation and retrieval in isolation without needing to pretrain new models. For example, RAG can be used to finetune a GPT3 generator with a ColBERT retriever right away, which were released near NeurIPS submission time.
- From a scientific standpoint, re-using existing components allows us to make direct comparisons with previously proposed/analyzed and widely-used models like BART (for generation) and BERT/DPR (for retrieval). The model re-use lets us highlight the contribution of our proposed end-to-end training procedure, specifically by not re-inventing the wheel for individual retrieval and generation components.

**Brevity of descriptions of models (R1, R3)**  R1 suggested that "A figure or example about PAG-Sequence Model and PAG-Token Model is needed", and R3 mentions "description of the model is quite concise (due to space restrictions)". We will happily add further exposition and add more detail on the differences between RAG-Sequence and RAG-Token.

**No document encoder training experiments (R1)**  We focus on query-encoder finetuning due to its low compute cost and simplicity (the large document index does not need to be updated during training). We show this method's effectiveness compared to a fixed retriever. We consider document-encoder training to be out-of-scope here, but agree it is an interesting topic for future work that could potentially lead to gains albeit with significantly higher compute cost.

**More architectures and retrieval supervision (R3)**  R3 suggested we could compare RAG to memory-network style approaches, as well as ablations looking at joint supervision. We agree that these experiments are interesting for completeness, but we believe that our existing baselines are sufficient to justify the effectiveness of our contribution.

**More Generator baselines (R4)**  R4 suggested "it would be more interesting to compare RAG with GPT-2 or T5 models rather than BART". We use BART in the RAG models in our experiments as has been shown it performed well on a number of language tasks, outperforming similarly-sized T5 models (see BART paper). Since we use BART for RAG, a BART-only generator is the appropriate baseline to determine the effect of retrieval augmentation. Additional generator baselines would be interesting for completeness but we argue our existing experiments are sufficient to support our conclusions.

[Meta-Review · NeurIPS 2020]

This work proposed a system that uses the retrieval results of query to aid the generation of answers. The idea is generally natural and has been explored by quite some authors in various ways. The paper is clearly written, and I enjoyed reading it. I would see this work as a nice piece of work that combines several existing models in a neat although not strikingly novel or inspirational way. The major downside of the work is its novelty, but its strong empirical results and potential impact on practice are enough to support its acceptance by NeurIPS.